# High-Q-Factor Tunable Silica-Based Microring Resonators

**Yue-Xin Yin** [1], **Xiao-Pei Zhang** [2], **Xiao-Jie Yin** [3,4], **Yue Li** [1], **Xin-Ru Xu** [1], **Jun-Ming An** [3], **Yuan-Da Wu** [3], **Xiao-Ping Liu** [2] and **Da-Ming Zhang** [1,*]

1   State Key Laboratory of Integrated Optoelectronics, College of Electronic Science and Engineering, Jilin University, Changchun 130012, China; yxyin20@mails.jlu.edu.cn (Y.-X.Y.); liyue19@mails.jlu.edu.cn (Y.L.); xuxr20@mails.jlu.edu.cn (X.-R.X.)
2   National Laboratory of Solid State Microstructures and College of Engineering and Applied Sciences, Nanjing University, Nanjing 210093, China; DZ1634007@small.nju.edu.cn (X.-P.Z.); xpliu@nju.edu.cn (X.-P.L.)
3   State Key Laboratory of Integrated Optoelectronics, Institute of Semiconductors, Chinese Academy of Sciences, Beijing 100083, China; yinxiaojie@semi.ac.cn (X.-J.Y.); junming@semi.ac.cn (J.-M.A.); wuyuanda@semi.ac.cn (Y.-D.W.)
4   Shijia Photons Technology, Hebi 458030, China
*   Correspondence: zhangdm@jlu.edu.cn

**Abstract:** A high-Q-factor tunable silica-based microring resonator (MRR) is demonstrated. To meet the critical-coupling condition, a Mach–Zehnder interferometer (MZI) as the tunable coupler was integrated with a racetrack resonator. Then, 40 mW electronic power was applied on the microheater on the arm of MZI, and a maximal notch depth of about 13.84 dB and a loaded Q factor of $4.47 \times 10^6$ were obtained. The proposed MRR shows great potential in practical application for optical communications and integrated optics.

**Keywords:** optical devices; resonators; integrated optics; wavelength filtering devices





## 1. Introduction

Owing to the compact footprint and functional versatility, optical microresonators have attracted much attention in many fields, including optical filters [1], sensors [2,3], and nonlinear optics [4]. Microresonator devices with a high Q factor were designed and demonstrated on different material platforms, such as silicon-on-insulator (SOI) [5–7], silicon nitride (SiN) [8–10], indium phosphide (InP)-based planar lightwave circuits (PLCs) [11–13], polymer-based PLCs [3,14,15], and silica-based PLCs [16]. Among them, benefitting from low loss, miniaturization, scalability, and high fiber-coupling efficiency, silica-based PLCs devices show great potential in optics integration and commercial field [17–19]. Whispering gallery mode (WGM)-based silica microtoroid resonators [20,21] with ultrahigh Q factors of >$10^8$ are used in frequency combs [22] and Brillouin lasers [23]. However, till now, silica-based high Q factor microring resonators (MRRs), a great candidate for optical interconnection and wavelength division multiplexing systems, have been less demonstrated [16]. To achieve high Q factor and notch depth, MRRs are designed to work under the critical-coupling condition. However, the critical-coupling condition cannot be easily achieved due to the possible deviation of the fabrication process. Replacing fixed directional couplers with tunable couplers is an efficient method to meet the critical-coupling condition [12,15,24,25]. In this paper, a high-Q-factor silica-based MRR was designed and is experimentally demonstrated. To obtain high-notch-depth MRRs, an MZI structure is used to replace the conventional directional coupler. The critical-coupling condition is satisfied by changing the coupling coefficient. The notch depth of the transmission from the through port can be tuned from 1.12 to 13.84 dB. Lastly, a high loaded Q factor of $4.47 \times 10^6$ was acquired, which means that the propagation loss of the waveguides was about 0.11 dB/cm.

## 2. Structure and Design

To realize the compact device, we designed the tunable MRRs on the basis of the 2% Δ silica-based PLC platform. Both the width and thickness of the waveguides were 4 μm to realize single-mode propagation [16]. The structure of the tunable MRRs is shown in Figure 1. We replaced the directional coupler by the MZI structure. The coupling coefficient of the MZI is charged with electrical power applied on the MZI arm.

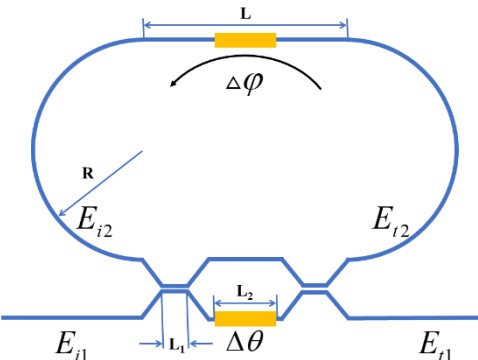

**Figure 1.** Structure of tunable microring resonator.

To decrease bent loss, we chose radius $R$ of the ring of 1600 μm. The length of the MZI arm was 3000 μm. Length $L_1$ of the two coupling regions of the MZI was 195 μm with a gap of 1.5 μm. Then, the gap was transferred from 1.5 to 127 μm using S-bent waveguides with a radius of 3000 μm. According to the MZI length, length $L$ of the straight waveguide was 7250 μm. On the basis of coupled-mode theory (CMT) and the transfer-matrix technique (TMT) [26,27], the normalized amplitude-response function of the tunable ring resonator is expressed as

$$P = |E_{t1}|^2 = \frac{\alpha^2 + t^4 + \kappa^4 - 2t^2\kappa^2\cos(\Delta\theta) - 2\alpha t^2\cos(\Delta\theta) + 2\alpha\kappa^2\cos(\Delta\theta)}{1 + \alpha^2\kappa^4 + \alpha^2 t^4 + 2\alpha\kappa^2\cos(\varphi + \Delta\varphi) - 2\alpha t^2\cos(\varphi + \Delta\varphi + \Delta\theta) - 2\alpha^2 t^2\kappa^2\cos(\Delta\theta)} \tag{1}$$

where $\kappa$ and $t$ are the coupling and transmission coefficient of a single directional coupler with the relationship of $\kappa^2 + t^2 = 1$; is the loss coefficient of the ring (zero loss $\alpha = 1$); $\varphi = L_{cir}\beta$ is the phase shift caused by the ring waveguide, $L_{cir}$ is the circumference of the ring, $\beta = \frac{2\pi}{\lambda}n_{eff}$ is the propagation constant with effective refractive index $n_{eff}$ and wavelength $\lambda$; $\Delta\varphi = L_2\frac{2\pi}{\lambda}\frac{dn}{dT}\Delta T$ and $\Delta\theta = L_2\frac{2\pi}{\lambda}\frac{dn}{dT}\Delta T$ are the phase shift caused by microheaters on the ring waveguide and the arm of MZI, respectively; $\frac{dn}{dT} = 1.19 \times 10^{-5}K^{-1}$ is the thermo-optic coefficient of $SiO_2$; the same length $L_2$ and width of the microheaters on the ring waveguide and the MZI arm are 2600 and 21 μm.

On the basis of the propagation loss of the silica-based waveguide that we calculated [19], we chose waveguide loss 0.1 dB/cm to optimize the parameters of the racetrack MRR. To simplify the calculation, the MZI inside the racetrack was assumed to be a straight waveguide with $L$. Hence, circumference $L_{cir} = 2\pi R + 2L$ was calculated as 2.455 cm. Then loss coefficient $\alpha$ was 0.9721. On the basis of Equation (1), we calculated the transmission of the tunable MRR through MATLAB. We changed the indices of the arm of the MZI and the ring waveguide under the microheaters to perform coupling-condition and resonator-wavelength tuning, respectively. Figure 2a shows the transmission with coupling-condition tuning. To perform the tunable effect, the MRR first worked in overcoupling condition with a $\kappa$ of 0.5315 according to the design parameters. With the change in the temperature of the MZI arm, the coupling condition was changed. The deepest depth of 25.04 dB was obtained. An FSR of 65.50 pm and ~4 pm/K wavelength shift was also observed. Figure 2b shows the normalized transmission of the resonator peak in critical-coupling condition while the temperature changes of 14 K. The simulated data shown by the blue hollow circles were fit using the theoretical Lorentzian transmission in Origin (see the solid red curve). Full width at half maximum (FWHM) of the resonance peak of 1533.17 nm for

the present resonator was about $\Delta\lambda = 0.0435$ pm, which indicated that a loaded Q factor $Q_{load}$ of $3.52 \times 10^7$ was obtained. Figure 3 shows transmission with resonator wavelength tuning. A wavelength shift of ~4 pm/K is observed.

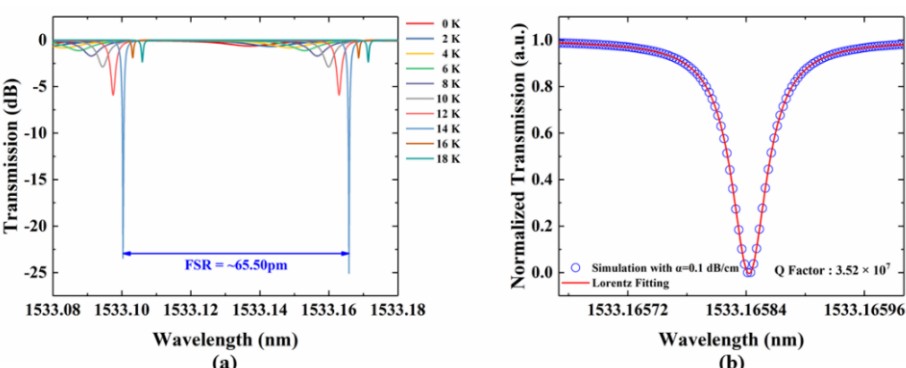

**Figure 2.** (**a**) Calculated normalized transmission with coupling-condition tuning of tunable MRR; (**b**) calculated deepest resonant peak with Lorentzian fitting.

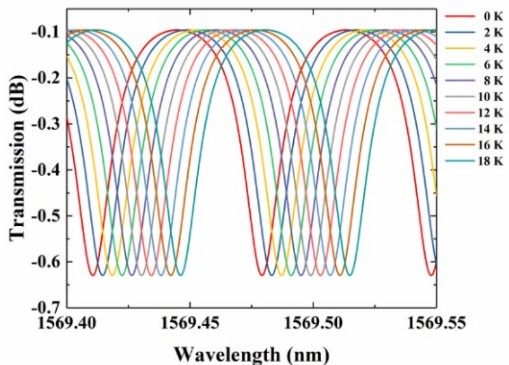

**Figure 3.** Calculated normalized transmission with resonant wavelength tuning of tunable MRR.

## 3. Fabrication and Measurement

The designed MRRs were fabricated by the PLC foundry, SHIJIA, China. The bottom cladding of ~15 μm was thermally oxidized on silicon substrate. A 4 μm thick core layer was fabricated through plasma-enhanced chemical-vapor deposition (PECVD). Then, the device was annealed above 1100 °C. We patterned the waveguides through ultraviolet (UV) lithography and fully etched the core layer by an inductively coupled plasma (ICP) dry-etching process with a $C_4F_8/SF_6$ gas mixture. Next, the top cladding of ~15 μm was deposited by PECVD and again annealed. Lastly, microheaters were deposited by means of magnetron sputtering. Figure 4a shows the tunable MRR photograph. The cross-section of the core waveguide is shown in Figure 4b. The geometry of the core waveguide was not exactly a square due to the fabrication process, increasing propagation loss and slightly shifting the resonant wavelength. Both the width and thickness of the core waveguides were smaller than 4 μm, ensuring single-mode propagation.

Because of the polarization sensitivity of the microring resonator, polarization-maintaining fiber (PMF) was used to couple the input and output ends of the chip, which were fixed by ultraviolet (UV) glue. Wire bonding between the heaters and printed circuit boards (PCB) finalized the chip-assembly procedure. To measure the fabricated MRR, we used a swept-wavelength laser source (New Focus TLB-6600) that was tunable from 1510 to 1590 nm. The output signal of the MRR was detected by a photodetector (Thorlabs PDA10CS, 17 MHz bandwidth) associated with a data-acquisition card (DAQ, National Instruments USB-6366). The measured transmissions were normalized with respect to the transmission from laser source to photodetector. Owing to the present resonator having a high Q factor, the step size of the tunable laser source was set to be 0.03 pm.

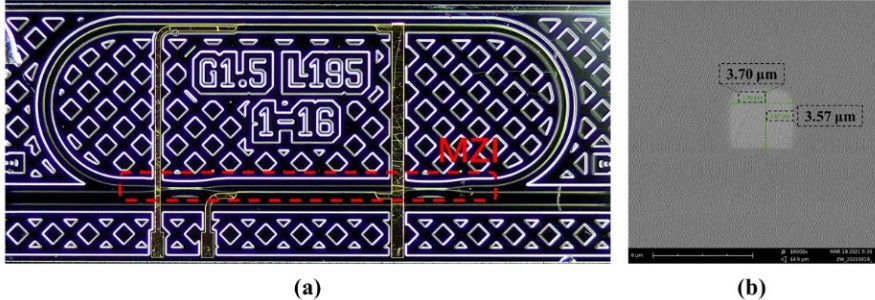

**Figure 4.** (**a**) Photograph of silica-based tunable MRR; (**b**) scanning electron microscope (SEM) image of core waveguide cross-section.

Both coupling condition and resonant wavelength were tunable by applied electric power using source meter instruments (Keithley 2450) to the microheaters on the MZI arm and the ring waveguide, respectively. As shown in Figure 5a, the extinction ratio (ER) of the MRR could be tuned by the applied power corresponding to the coupling coefficient tuning of the MZI-based coupler. The insertion loss (IL) of our MRR chip was ~3.72 dB, including the coupling loss between the fibers and the waveguides, and the insertion loss of the tunable MRR. At the beginning, the ring resonator was working at the overcoupling condition, as most of the light in the input waveguide was coupled into the ring resonator after passing the MZI. Hence, a shallow dip of around ~1.12 dB was observed. The light coupled into the ring resonator was decreased by increasing the applied power. When this coupled energy was equal with the round-trip loss of the ring, the critical-coupling condition was met, while the electrical power applied on the arm of MZI was 40 mW; the highest ER (about 13.84 dB) was obtained. An FSR of ~63.89 pm is shown in Figure 5a. Figure 5b shows the normalized transmission of the resonator peak in a critical-coupling condition. The measured data shown by the blue hollow circles were fitted using the theoretical Lorentzian transmission in Origin (see solid red curve). The FWHM of the resonance peak 1533.16 nm for the present resonator was about $\Delta\lambda = 0.343$ pm, which indicated that a loaded ultrahigh-Q factor $Q_{load}$ of $4.47 \times 10^6$ was obtained. Since the device operated under a critical-coupling condition, we considered $Q_{load}$ to be half of intrinsic Q factor $Q_{int}$ [5]. Therefore, propagation loss of 0.11 dB/cm of the silica-based waveguide was obtained by using equation $Q_{int} \approx 2\pi n_g / \lambda\alpha$ [5], where $\alpha$ is the propagation loss, and the group index is ~1.4615. A deviation of the core geometry affected the effective refractive index of the waveguide. The introduced phase caused the resonance peak shift. The propagation loss of the waveguide was also 0.01 dB/cm larger than that of the simulation. The main reason may have been the loss caused by the MZI. The continued increasing power forced the ring to work in an undercoupling condition. A small resonant wavelength shift was generated during the coupling-condition tuning process because of the additional phase shift of the MZI-based coupler to the ring resonator. When the ring resonator was used as a filter, the small resonant wavelength shift was compensated with the microheater on the ring waveguide. Figure 6 shows the wavelength shift caused by coupling-condition tuning with 3.956 pm/mW.

The resonant wavelength could be tuned by applying electric power to the heater on the ring waveguide. As shown in Figure 7a, owing to the positive thermo-optic coefficient of SiO$_2$ [28], a red shift could be observed. The blue circles mark the resonance peak under the same order. The relationship between applied power and resonant wavelength shift is shown in Figure 7b with a tuning efficiency of ~4.136 pm/mW.

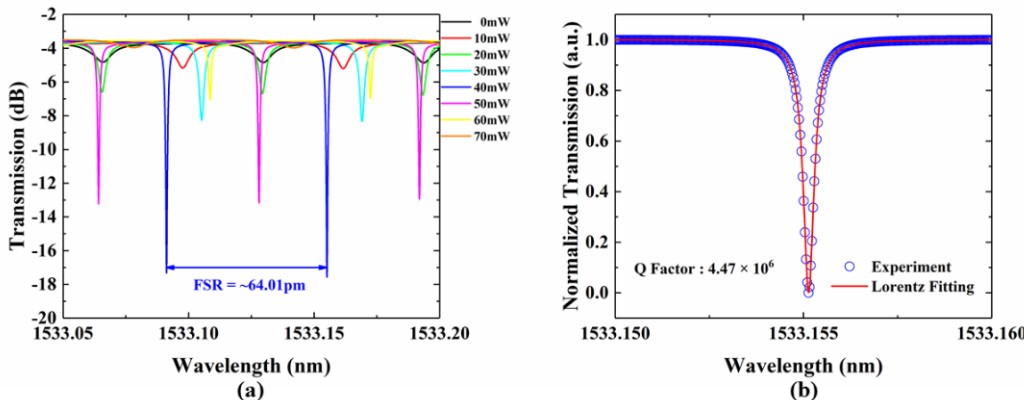

**Figure 5.** (**a**) Coupling-condition tuning of tunable MRR; (**b**) deepest resonant peak with Lorentzian fitting.

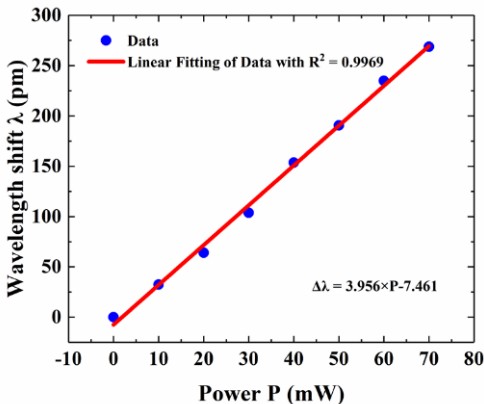

**Figure 6.** Resonant wavelength shift caused by coupling-condition tuning.

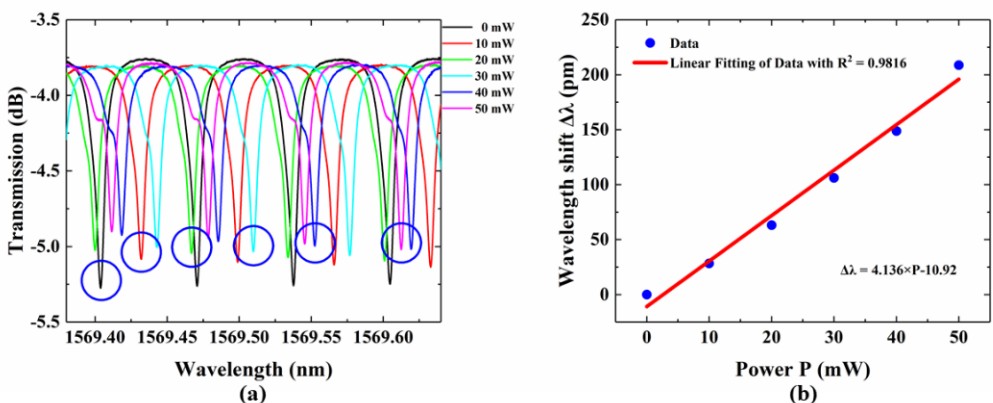

**Figure 7.** (**a**) Resonant-wavelength tuning of tunable MRR; (**b**) relationship between resonant-wavelength shift and electronic power.

## 4. Discussion

Table 1 shows a comparison of the microring resonators reported in recent years based on different platforms. The fabrication process and structure were optimized to improve performance of MRRs. InP-based PLC offers a hybrid integration platform with active and passive devices. Through applying tunable structure, the Q factor of InP-based PLC was raised up to ~$10^4$ [11,12]. To achieve a high-Q-factor InP MRR, a large radius of 13 mm was used in [13]. A 3 inch InP wafer could fabricate four such devices. However, the Q factor was still lower than the MRR proposed in this paper. Limited by the loss of waveguides, the tunable polymer-based MRR [15] obtained a Q factor of $8.2 \times 10^4$. Besides the small size

of waveguides, the fiber-to-chip coupling efficiency deteriorated to ~15 dB [15]. Benefitting from the high refractive index contrast between silicon core and silica cladding, the SOI MRR could be more compact than ever. In [7], an ultrahigh Q factor of $1.3 \times 10^6$ and a large FSR of 900 pm were obtained by a bent directional coupler. However, the device showed a lack of tolerance and low ER. On the other hand, a large radius decreased the loss but also lowered FSR [29], which limits its application. The same situation happened in SiN MRR. To achieve an ultrahigh Q factor, optimized ICP and chemical mechanical polishing (CMP) are utilized to reduce sidewall and surface roughness [9]. Though a $3.7 \times 10^7$ Q factor was obtained, the fabrication process is more complex. An ultrahigh Q factor of $4.22 \times 10^8$ SiN MRR was demonstrated in [10]. The ultralarge radius of 11.789 mm limits its application. Compared with our previous work [16], both ER and Q factor were improved. The proposed tunable silica-based MRRs were fabricated by the PLC foundry on a 6 inch silica-on-silicon wafer for commercial use. With a tunable structure, the devices overcame the deviation of the fabrication process. A high Q factor of $4.47 \times 10^6$ could be obtained while the ER was 13.84 dB, which means that the propagation loss of the waveguide was 0.11 dB/cm. Though the FSR of 64 pm was small, it could be improved by cascading MRRs.

**Table 1.** Comparison of high-Q MRRs.

| Ref. | Platform | R (μm) | FSR (pm) | ER (dB) | Q Factor |
|------|----------|--------|----------|---------|----------|
| [11] | InP-based PLC | 200 | 250 | 10 | $2.2 \times 10^4$ |
| [12] | InP-based PLC | 50 | N.A. | 18.5 | N.A. |
| [13] | InP-based PLC | 1300 | 17.8 | 7.0 | $0.97 \times 10^6$ |
| [15] | Polymer-based PLC | 500 | 130 | 18 | $8.2 \times 10^4$ |
| [7] | SOI | 29 | 900 | 4 | $1.3 \times 10^6$ |
| [29] | SOI | 2600 | 5.1 GHz | 8 | $7.5 \times 10^7$ |
| [9] | SiN | 115 | N.A. | N.A. | $3.7 \times 10^7$ |
| [10] | SiN | 11,787 | 2.713 GHz | N.A. | $4.22 \times 10^8$ |
| [16] | Silica-based PLC | 1600 | 137 | 3 | $1.83 \times 10^6$ |
| This Work | Silica-based PLC | 1600 | 64 | 13.84 | $4.47 \times 10^6$ |

## 5. Conclusions

A high-Q-factor tunable MRR was well-designed, fabricated, and experimentally demonstrated. The ER of the MRR could be tuned from 1.12 to 13.84 dB. A high Q factor of $4.47 \times 10^6$ was obtained while the ER was 13.84 dB, which means that the propagation loss of the waveguide was 0.11 dB/cm. The MRR was fabricated by the silica-based PLC foundry. The MRR was optically and electrically well-packaged. The proposed MRR shows great potential in practical application for optical communications and integrated optics.

**Author Contributions:** Methodology, Y.-D.W.; software, Y.-X.Y.; validation, Y.L. and X.-R.X.; formal analysis, X.-P.Z. and X.-P.L.; investigation, Y.-X.Y., Y.L. and X.-R.X.; resources, J.-M.A. and Y.-D.W.; data curation, Y.-X.Y., X.-P.Z. and Y.L.; writing—original-draft preparation, Y.-X.Y. and Y.L.; writing—review and editing, D.-M.Z.; project administration, D.-M.Z.; funding acquisition, X.-J.Y. and D.-M.Z.; All authors have read and agreed to the published version of the manuscript.

**Funding:** This research was funded National Key Research and Development (R&D) Program of China (2019YFB2203004), and the Science and Technology Development Plan of Jilin Province (20190302010GX).

**Institutional Review Board Statement:** Not applicable.

**Informed Consent Statement:** Not applicable.

**Data Availability Statement:** Not applicable.

**Conflicts of Interest:** The authors declare no conflict of interest.

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
