# Peer review of "High-Q-Factor Tunable Silica-Based Microring Resonators"

_photonics, doi:10.3390/photonics8070256_

Round 1

Reviewer 2 Report

The manuscript describes an experimental study of the silica-based microring resonator. It includes describing the modeling, fabrication process, and experimental testing. The Mach-Zander interferometer was used as a coupler and allowed to achieve tunable coupling, which can be interesting for some applications. Q-factor of 4.47 million at the critical coupling is reported. A dry etching process was used for manufacturing the chip, which is promising for mass production.

In the introduction, the Q in silica resonators wasn’t discussed. The articles about ultra-high Q-factors in an on-chip silica microresonator should be mentioned.

  1. Lue Wu, Heming Wang, Qifan Yang, Qing-xin Ji, Boqiang Shen, Chengying Bao, Maodong Gao, and Kerry Vahala, "Greater than one billion Q factor for on-chip microresonators," Opt. Lett. 45, 5129-5131 (2020)

  1. Lee, H., Chen, T., Li, J. et al. Chemically etched ultrahigh-Q wedge-resonator on a silicon chip. Nature Photon 6, 369–373 (2012). https://doi.org/10.1038/nphoton.2012.109

In the Structure and Design section, the response function of the resonator was introduced, but there is no description of where it was used and what for. The coefficient t is not explained in the text. The paragraph ended with ; . Figures 2 and 3 insufficiently described in the text, how it was calculated stayed unclear. Was the (1) used for it? How was the frequency shift due to temperature change calculated?

In fig. 4(a) in the Fabrication and Measurement section it would be nice to add a scale. Is the microring single-mode? Does the coupling monotonously increase in the fig. 5(a) with the electric power increase? It seems to me that 70 mW corresponds to the critical coupling, but this is not directly stated in the text. Why is the coupling efficiency different for critical coupling for different modes? The colors in the fig 5(a) and 7(a) do not contrast to distinguish the lines intuitively. Something marked with circles in the fig. 7(a) but it isn't mentioned in the text. 

In the discussion, the bending losses were mentioned as a limiting factor. It might be useful to make estimations on the losses for described design. Is Q close to that limit? Until that, for me, the novelty of the manuscript seems doubtful due to the results in the articles mentioned above.

There are many mistakes and incomprehensible sentences along with the text, which complicates the perception of information and spoils the impression of the work. The quality of the result presentation leaves much to be desired. 

Reviewer 3 Report

This is a very good article, scientifically sound and technically interesting.  The article claim the demonstration of a high Q factor tunable silica-based microring resonator (MRR) Also that while 40mW electronic power applied on the microheater on the arm of MZI, the maximum notch depth of about 13.84 dB and a loaded Q factor of 4.97x 10(6) are obtained.

Was the affect of the microheater systematically studied on the maximum notch depth and Q factor.  Could you increased or diminished by varying the microheater power.  What about the temperature of the device? Often this is a more important factor since heat dissipation through conduction in the device itself, can widely change the effect of any heater.

How reliable is the microheater? If you disconnect or remove the microheater what maximum notch depth and Q did you obtained?

It is mentioned that the critical coupling condition is satisfied through changing the coupling coefficient.  The coupling coerficient of the MZI is changed with voltages applied on the MZI arm.  Could you provide more technical detail?

Reviewer 4 Report

The authors propose and demonstrate a silica micro-resonator using a tunable MZI to control the coupling rate, thus the coupling condition to the ring. Based on this approach, the authors demonstrate experimentally critical coupling with an extinction ratio of 13.84 dB and a quality factor of 4.47 x 106. I consider this work to be interesting for the community and worth being published in Photonics, provided the following important issues are properly addressed:

Comment 1:

I think the contextualization of this work in comparison with the state of the art should be improved. Is it the first time a tunable MZI is used to tune the coupling conditions of a PLC silica-based micro-ring resonators? The performance of this work should be compared with previous demonstrations from other groups using similar approach in silica.

Comment 2:

If I well understood, when calculating propagation loss using the formula Qint = (2  ng) / (lambda alfa), the authors consider that alfa is due to propagation loss only. What about loss in the MZI? Do the authors have any experimental data validating that the loss in MZI is low enough to neglect it in the loss calculation?

Comment 3:

Has the use of the MZI any influence in the quality factor of the resonator? In other words, how does the Q of this tunable resonator compares with that of conventional resonators made in the same technology?  

Comment 4:

The authors mention that “The cross section of the core waveguide is shown in Figure 4 (b). The geometry of core waveguide is not exactly a  square due to the fabrication process.” It would be interesting if the authors could elaborate a bit more this point. Is it due to annealing process, etching, a combination of both?

Comment 5:

The following sentence may be a bit confusing : “the device is annealed above 1100 ℃ to be compact” The term compact may be understood related to the size, e.g. a very compact resonator with a size of only… I would propose to rephrase it.

Author Response

Response Letter

Manuscript ID: 1263962

Paper Type: Article

Title: High-Q-Factor Tunable Silica-Based Microring Resonators

Authors: Yue-Xin Yin, Xiao-Pei Zhang, Xiao-Jie Yin, Yue Li, Xin-Ru Xu, Jun-Ming An, Yuan-Da Wu, Xiao-Ping Liu and Da-Ming Zhang

Dear Editor:

Thank you very much for your favorable decision on our manuscript. We are very grateful to the reviewers for their careful reviews and valuable suggestions, which have helped us improve the overall quality of our manuscript. Our responses to the reviewers' comments and the summary of the changes are listed below.

We hope this revision will meet the publication criterion of Photonics. If there are any further issues, please don't hesitate to let us know. Many thanks to you for your time and effort on our manuscript.

Sincerely yours,

Yue-Xin Yin,

Da-Ming Zhang,

College of Electronic Science and Engineering, Jilin University,

Changchun 130012, China

Response to Reviewer 4

The authors propose and demonstrate a silica micro-resonator using a tunable MZI to control the coupling rate, thus the coupling condition to the ring. Based on this approach, the authors demonstrate experimentally critical coupling with an extinction ratio of 13.84 dB and a quality factor of 4.47 × 106. I consider this work to be interesting for the community and worth being published in Photonics, provided the following important issues are properly addressed:

Response: We would like to thank the reviewer for his/her inspiring comments and constructive suggestions, which have allowed us to improve our manuscript. We have corrected our text as per his/her comments, and our responses to the detailed concerns are listed as follows.

Comment #1: I think the contextualization of this work in comparison with the state of the art should be improved. Is it the first time a tunable MZI is used to tune the coupling conditions of a PLC silica-based micro-ring resonators? The performance of this work should be compared with previous demonstrations from other groups using similar approach in silica.

Response: Thanks for your valuable comment. The micro-resonator based on silica-on-silicon platform fabricated as microtoroid [R1] or microspheres [R2] with Q factor >108 for lasers and frequency combs have been demonstrated. To the best of my knowledge, it’s the first time silica based tunable MRR demonstrated in this paper. Hence, we compare our device with other platforms, such as InP [25] and Polymer [20] in Discussion section of this paper. Limit by the waveguide loss of InP and polymer based waveguides, the Q factor is only ~104 when critical coupling condition is met. Therefore, our silica based tunable MRR shows great potential in optical interconnection and wavelength division multiplexing system.

Action taken: We have added two sentences, " Especially, whispering gallery mode (WGM) based silica microtoroid resonators with Q factors of >108 have been used in frequency combs and Brillouin laser. However, till now, silica based high Q factor microring resonators (MRRs), a great candidate for optical interconnection and wavelength division multiplexing system, are less demonstrated." in the first paragraph from line 32 to 36 on page 1 in the manuscript to state that the comparison between microtoroid resonators and microring resonators.

Comment #2: If I well understood, when calculating propagation loss using the formula Qint = (2ng) / (lambda alfa), the authors consider that alfa is due to propagation loss only. What about loss in the MZI? Do the authors have any experimental data validating that the loss in MZI is low enough to neglect it in the loss calculation?

Response: We thank the reviewer for bringing up this inspiring issue. The simulation of the MZI is shown in Figure 1. The excess loss of MZI is 0.096 dB (0.978=0.936+0.042). The length of the MZI is 6852μm. Assuming the propagation loss is 0.1dB/cm [13]. The total loss caused by MZI is 0.068 dB. There is a difference value 0.028dB between assumed loss and actual loss. The circumference of the racetrack is 2.455 cm ( ). Hence the total loss of the racetrack is 2.455 dB. Compared with the difference value (2.455dB>>0.028dB), the loss can be neglected. However, to improve Q factor further, this loss should be considered.

Figure 1 Simulation of MZI

Comment #3: Has the use of the MZI any influence in the quality factor of the resonator? In other words, how does the Q of this tunable resonator compare with that of conventional resonators made in the same technology?

Response: We thank the reviewer for bringing up this inspiring issue. The all-pass MRR based on the same technology has been demonstrated in [R.3]. Both of MRRs have a Q factor of >106. However, suffering deviation of fabrication process, the all-pass MRR is hard to work under critical coupling condition. The proposed MRR with coupling tuning could achieve critical coupling easier and larger Q factor of 4.47×106. To further improve the tunable MRR performance, we believe the loss inside the MZI should also be considered. Thanks again for your valuable comment.

Figure 2 (a) Part of the measured transmission spectrum. (b) One peak of measured transmission spectrum with Lorentzian fitting [R2]

Comment #4: The authors mention that “The cross section of the core waveguide is shown in Figure 4 (b). The geometry of core waveguide is not exactly a square due to the fabrication process.” It would be interesting if the authors could elaborate a bit more this point. Is it due to annealing process, etching, a combination of both?

Response: We thank the reviewer for bringing up this inspiring issue. The change of the geometry of core waveguide is caused by many factors. Not only annealing and etching, but also ultra-violet (UV) lithography position on the wafer. Therefore, we summarize them as the possible deviation of fabrication process.

Comment #5: The following sentence may be a bit confusing : “the device is annealed above 1100 ℃ to be compact” The term compact may be understood related to the size, e.g. a very compact resonator with a size of only… I would propose to rephrase it.

Response: We thank the reviewer for his/her constructive suggestion. We have corrected these mistakes in the revised manuscript.

Action taken: We have deleted “to be compact”.

Reference:

  • Lee, H.; Chen, T.; Li, J.; Yang, K.Y.; Jeon, S.; Painter, O.; Vahala, K.J., Chemically etched ultrahigh-q wedge-resonator on a silicon chip. Nature Photonics 2012, 6, 369-373.
  • Maker, A.J.; Armani, A.M., Fabrication of silica ultra high quality factor microresonators. J Vis Exp 2012.
  • Yin, Y.-X.; Yin, X.-J.; Zhang, X.-P.; Yan, G.-W.; Wang, Y.; Wu, Y.-D.; An, J.-M.; Wang, L.-L.; Zhang, D.-M., High-q-factor silica-based racetrack microring resonators. Photonics 2021, 8.

List of changes

We have revised the manuscript according to the suggestions of the reviewers. The changes we have made are as follows:

  1. We have changed “micro-ring resonators” to “optical micro-resonators” in Line 24 on Page 1.
  2. We have changed “MRR-based” to “Micro-resonators” in Line 26 on Page 1.
  3. We have added some reference from Line 27 to 29 on Page 1.
  4. We have changed “Moreover, ultralow loss silica waveguide is an ideal platform for ultrahigh Q factor MRRs.” to “Especially, whispering gallery mode (WGM) based silica microtoroid resonators [14,15] with Q factors of >108 have been used in frequency combs [16] and Brillouin laser [17]. However, till now, silica based high Q factor microring resonators (MRRs), a great candidate for optical interconnection and wavelength division multiplexing system, are less demonstrated [10]” from Line 32 to 36 on Page 1.
  5. We have changed “drop” to “through” in Line 44 on Page 1.
  6. We have changed “voltage” to “electrical power” in Line 54 on Page 2.
  7. We have added “where and  are coupling and transmission coefficient of a single directional coupler with the relationship of  ;” from Line 63 to 64 on Page 2.
  8. We have added is the propagation constant with the effective refractive index  and the wavelength ;  and  are the phase shift caused by micro-heaters on the ring waveguide and the arm of MZI, respectively;  is the thermo-optic coefficient of SiO2;” from Line 66 to 69 on Page 2.
  9. We have added “To simplify the calculation, the MZI inside the racetrack is assumed to be a straight waveguide with . Hence, the circumference is calculated as 2.455 cm.” from Line 73 to 74 on Page 2.
  10. We have added “Based on Equation (1), we calculate the transmission of the tunable MRR through MATLAB. We change the indices of the arm of the MZI and the ring waveguide under the micro-heaters to perform coupling condition tuning and resonator wavelength tuning, respectively.” from Line 75 to 78 on Page 2.
  11. We have deleted “to be compact” in Line 100 on Page 3.
  12. We have added “Both of width and thickness of core waveguides are smaller than 4μm ensuring single mode propagation.” From Line 108 to 109 on Page 3.
  13. We have changed “an ultra-high Q factor” to “a high Q factor” in Line 121 on Page 4.
  14. We have added “while the electrical power applied on the arm of MZI is 40mW” from Line 133 to 134 on Page 4.
  15. We have added “The deviation of the core geometry will affect the effective refractive index of the wave-guide. Then the introduced phase will cause the resonance peak shift. Propagation loss of waveguide is also 0.01 dB/cm large than simulation. The main reason should be the loss caused the MZI.” from Line 144 to 147 on Page 4.
  16. We have changed the colors of the lines in Figure 5(a) and Figure 7(a).
  17. We have added “The blue circles mark resonance peak under the same order.” From Line 164 to 165 on Page 5.
  18. We have rephrase the discussion section “InP based PLC offers a hybrid integration platform with active and passive devices. Though applying tunable structure, the Q factor of InP based PLC is up to ~104 [11,12]. To achieve a high Q factor InP MRR, a large radius of 13mm is used in Ref. [13]. A 3-inch InP wafer can fabricate four such devices. However, the Q factor is still lower than the MRR proposed in the paper. Limited by the loss of waveguides, the tunable polymer based MRR [15] obtains a Q factor of 8.2 × 104. Beside small size of waveguides, the fiber-to-chip coupling efficiency deteriorates to ~15 dB [15]. Benefit from high refractive index contrast between the silicon core and the silica cladding, SOI MRR could be compact than ever. In Ref. [7], an ultrahigh-Q of 1.3 × 106 and large FSR of 900pm are obtained by a bent directional coupler. However, the device shows lack of tolerance and low ER. On the other hand, large radius decrease the loss but also lower FSR[29], which will limit its application. The same situation happens in SiN MRR. To achieve ultra-high Q factor, optimized ICP and chemical mechanical polishing (CMP) are utilized to reduce sidewall and surface roughness [9]. Though 3.7 × 107 Q factor is obtained, the fabrication process will be more complex. An ultra-high Q factor of 4.22 × 108 SiN MRR is demonstrated in Ref. [10]. The ultra-large radius of 11.789mm will limit its application.” from 170 to 185 on Page 5 and 6.
  19. We have added literatures and changed the order in Table 1.

Round 2

Reviewer 2 Report

Authors improved the quality of the presentation of their results. It became clear what has been done. I am ok with that.